# Soil Management in Indigenous Agroforestry Systems of Guarana (*Paullinia cupana* Kunth) of the Sateré-Mawé Ethnic Group, in the Lower Amazon River Region

Clara Peres Vignoli [1], Johannes Leeuwen [1], Robert Pritchard Miller [2], César Augusto Ticona-Benavente [1], Bárbara Vieira da Silva [1], Bruno Striffler [1], José Guedes Fernandes Neto [1,3] and Sonia Sena Alfaia [1,*]

1   National Institute of Amazonian Research, Manaus 69067-375, Brazil
2   Instituto Sociedade, População e Natureza (ISPN), Brasilia 70750-515, Brazil
3   Applied Ecology Graduate Program, University of São Paulo, São Paulo 05508-060, Brazil
*   Correspondence: sonia.alfaia@inpa.gov.br

**Abstract:** The Sateré-Mawé Brazilian indigenous people cultivate the guarana liana in biodiverse agroforests that incorporate many species, mainly trees, and produce food, medicines, fuel and income. The objective of this study was to evaluate the soil fertility in nineteen indigenous guarana agroforestry systems (AFSs) compared with areas of adjacent forests in the Andirá–Marau Indigenous Land—an ancestral territory of the Sateré-Mawé people. Soils in both locations showed low natural fertility, with levels of most macronutrients below the minimum agronomic reference levels. Higher C and soil organic matter (SOM) content was observed in the forest soils and may be related to greater vegetation cover and higher litter production. However, the maintenance of the average levels of most nutrients in the soil of the AFSs, at the same level as under the forests, suggests that efficient nutrient cycling is taking place. In these conditions, the management of organic matter seems to be essential to maintain the productivity of guarana AFSs since levels of C, SOM and organic N were not considered as low as those of most of the other soil attributes that were evaluated. The high tree diversity in the guarana agroforests, including N-fixing species, may also contribute to efficient nutrient cycling and maintenance of the soil food web. The results suggest that the addition of limestone, green manure practices and the reincorporation of organic residues from guarana processing, among other sources, can be a low-cost alternative to improve soil fertility and increase guarana production in the AFS under study.

**Keywords:** organic matter management; nutrient cycling; ethnodevelopment; Brazilian Amazon

## 1. Introduction

Guarana (*Paullinia cupana* Kunth) is a wild liana that was domesticated by the Sateré-Mawé people and is central to their mythology and cultural identification. More recently, it has been transformed into a plant of economic and therapeutic importance, whose caffeine-rich seeds are used both in soft drinks and syrups as well as for medicinal purposes [1]. The first historical record of guarana was in 1669, by Jesuit priest João Felipe Bettendorf [2]. In the year 1757, Jesuit priest João Daniel described guarana also as a component of the diversified homegardens in villages of the Sateré-Mawé people in the region of the Lower Amazon River [3]. Brazil is the only country in the world to produce guarana on a commercial scale, and practically all national production is consumed by the domestic market. In 2019, the national production reached 2761 tons [4] of which about 70% is absorbed by the soft drink industry, while the remaining 30% is sold as sticks of concentrated powder, syrup, powder or extract for export and for the pharmaceutical industry [5]. Currently, Sateré-Mawé's guarana, known for its pharmacological and therapeutic characteristics [6], is exported to European countries such as France and Italy.

Guarana is an allogamous plant, pollinated mainly by bees and with $2n = 210$ chromosomes, while seven other *Paullinia* species present $2n = 24$ [7]. The Sateré-Mawé traditionally manage the plant's genetic diversity by collecting seedlings for new plantings in the forests under wild mother plants, which makes their territory the largest in situ genetic bank of guarana [8]. Guarana is cultivated in biodiverse agroforestry systems (AFSs) that incorporate many species, mainly trees, which produce food, medicine, fuel and income [9]. Guarana AFSs are an example of the indigenous agroforestry traditions that most likely evolved, along with the domestication of forest species, from the dawn of agriculture in the region and are part of an important cultural heritage still maintained by many tribes in the Amazon [10]. Agroforestry as a land-use system can combine food security, income creation and soil fertility maintenance [11–13] and in the Amazon can play a fundamental role in the conservation of natural resources [14].

The present work is part of a larger project, called Waraná (guarana in the Sateré-Mawé language), which was sponsored by the Petrobras Socioenvironmental Program, developed between 2014 and 2017 with the communities of the Andirá–Marau Indigenous Territory, supporting sustainable farming practices with an emphasis on agroecology. The objective of this study was to evaluate the fertility of soils in indigenous agroforestry systems in guarana compared to adjacent primary forest areas.

## 2. Materials and Methods

### 2.1. Characterization of the Study Site

The Andirá–Marau Indigenous Land (TI–Terra Indígena), ancestral territory of the Sateré-Mawé people, covers an area of 788,528 hectares and was formally recognized in 1986 [15]. It is located in the center of the Brazilian Amazon region, with a part in the state of Para and part in the state of Amazonas (Figure 1). It overlaps two conservation areas: the Pau-Rosa National Forest (3% overlap of TI) and the Amazon National Park (11% overlap of TI). A total of 15,776 individuals reside in the territory, with 8037 living on the Andirá River, 6599 on the Marau River and 1140 on the Waicurapá River [16]. TI Andirá–Marau is located in a tropical forest environment, and its rivers and streams are of clear or dark water, indicative of an ancient geological substrate and low fertility soils. The climate is "Am" in the Köppen–Geiger classification, in which the average temperature of the coldest month is above 18 °C, the annual precipitation is above 1500 mm, and there is a dry period with monthly rainfall below 60 mm.

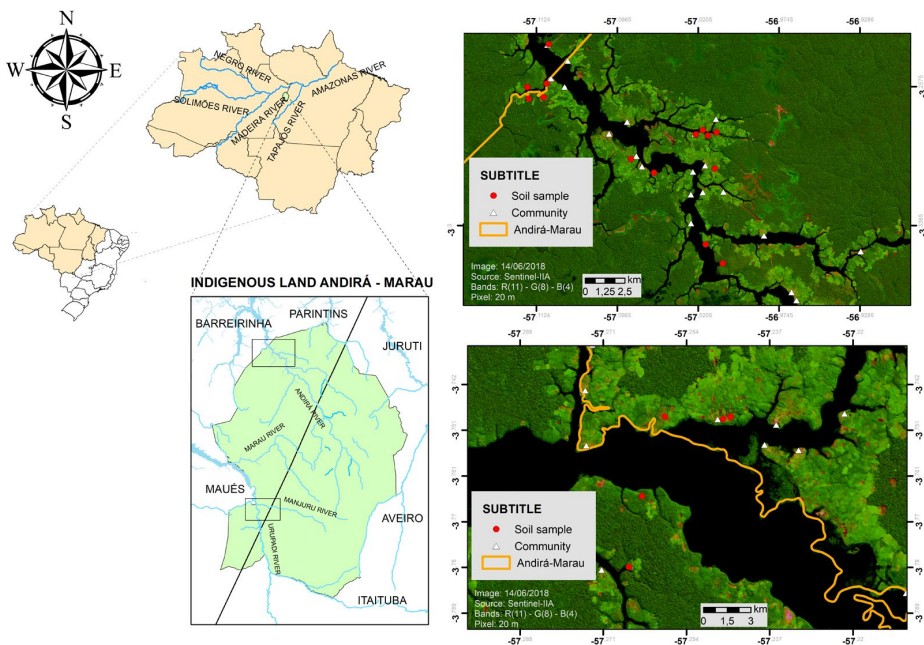

**Figure 1.** Location of the Andirá–Marau Indigenous Land in the Lower Amazon River region. Left above: the state of Amazonas. Left below: the Andirá–Marau Indigenous Land. Above right: communities along the Andirá River. Below right: communities along the Marau River.

### 2.2. Selection of the AFS Study Sites and Soil Sampling

Nineteen guarana agroforests were selected, with ages varying from 10 to 60 years. Fourteen were located in the Andirá River valley and five in the Marau River valley. The area of the AFS varied from 0.3 to 3 ha, with most below 1 ha in size. In the center of each AFS, an area of 30 m × 45 m was marked, containing three plots of 15 m × 30 m. In each plot, seven soil samples were collected in the 0–10, 10–20 and 20–30 cm layers, and these were mixed to obtain a single sample for each layer. Soil under adjacent primary forests, with no history of agricultural use in the last 40 years, was sampled as a control.

### 2.3. Analysis of the Soil Samples

The composite samples were analyzed at the Instituto Nacional de Pesquisas da Amazônia's (INPA) Thematic Soil and Plants Laboratory. The arithmetic mean of the triplicate plots represented the sample unit. Chemical analyses were carried out according to the methodology proposed by Embrapa [17]. The pH in $H_2O$, the exchangeable levels of Ca, Mg, K, Al and available P, organic carbon (C), nitrogen (N) and micronutrients Zn, Mn and Fe were determined. Soil pH in $H_2O$ was determined in a soil/water ratio 1:2.5, and samples were read in potentiometer. The exchangeable cations Ca, Mg and Al were extracted with KCl, and N, P, K, Zn, Mn and Fe were extracted with double acid (0.0125 M $H_2SO_4$ + 0.05 M HCl). Organic C was determined by the Walkley–Black method and nitrogen by Kjeldahl [17]. Organic matter was calculated from the organic C content.

The proportions of sand, silt and clay in the soil were determined by the rapid dispersion method [17]. Only the portion of the soil with particles less than 2 mm in diameter was considered, with sand (0.06 and <2), silt (0.002 and <0.06) and clay (<0.002).

### 2.4. Data Analysis

The statistical significance of the difference between the soil parameters of the guarana areas and under the forests was tested with Student's *t* test. To visualize relationships between soil fertility parameters, biplot graphs were constructed through a principal component analysis (PCA) with the standardized averages of the physical and chemical parameters. The software used to perform the analyses was SAS 9.0 (NY, EUA) and JMP 10 [18].

## 3. Results

### 3.1. Soil Chemical Attributes

The results of the chemical analysis show that there were no significant differences between the soils of the guarana AFSs and the primary forests in the values of pH, Ca, Mg, P, Mn and Fe (Table 1). The chemical characteristics that showed statistically significant differences when comparing the soils of the AFSs and forest (Al, K, SOM, C and N) are shown in Figure 2. The exchangeable Al saturation was higher in forest soils than in the AFSs, with significant differences at depths of 0–10 ($p < 0.05$) and 10–20 cm ($p < 0.01$). The contents ranged from medium (0.5 to 1.5 $cmol_c$ $kg^{-1}$) to high (>1.5 $cmol_c$ $kg^{-1}$) in the upper layer of the forest soil, according to the criteria of Cochrane et al. [19]. Exchangeable K and organic N in the soil showed significant differences between the two sampled systems, with the highest concentrations in the forest soils ($p < 0.05$) at all sampled depths. All samples showed organic carbon content to be average (8 to 26 g $kg^{-1}$) [19,20], with statistically significant, higher levels found in forest soils in the layers 0–10 cm ($p < 0.05$) and 10–20 cm ($p < 0.05$). Consequently, soil organic matter (SOM) also showed significant differences between the AFSs and nearby forest in the three soil layers.

**Table 1.** Values of pH, Ca, Mg, P, Mn and Fe at three soil depths in the guarana indigenous agroforestry systems and in the adjacent primary forests, in the Lower Amazon region ($n = 19$).

| Depth (cm) | pH (H₂O) | | Ca | | Mg | | P | | Mn | | Fe | | Zn | |
|---|---|---|---|---|---|---|---|---|---|---|---|---|---|---|
| | Forest | AFS | Forest | AFS | Forest | AFS | Forest | AFS | Forest | AFS | Forest | AFS | Forest | AFS |
| | | | | | $mol_c$ $kg^{-1}$ | | | | | | $mg$ $kg^{-1}$ | | | |
| 0–10 | 4.12 | 4.21 | 0.10 | 0.11 | 0.09 | 0.08 | 1.95 | 1.98 | 1.95 | 1.88 | 233 | 219 | 0.40 | 0.33 |
| 10–20 | 4.17 | 4.26 | 0.09 | 0.10 | 0.08 | 0.07 | 1.70 | 1.66 | 1.79 | 1.73 | 241 | 238 | 0.30 | 0.31 |
| 20–30 | 4.21 | 4.32 | 0.08 | 0.08 | 0.07 | 0.06 | 1.39 | 1.21 | 1.91 | 1.67 | 253 | 249 | 0.28 | 0.25 |

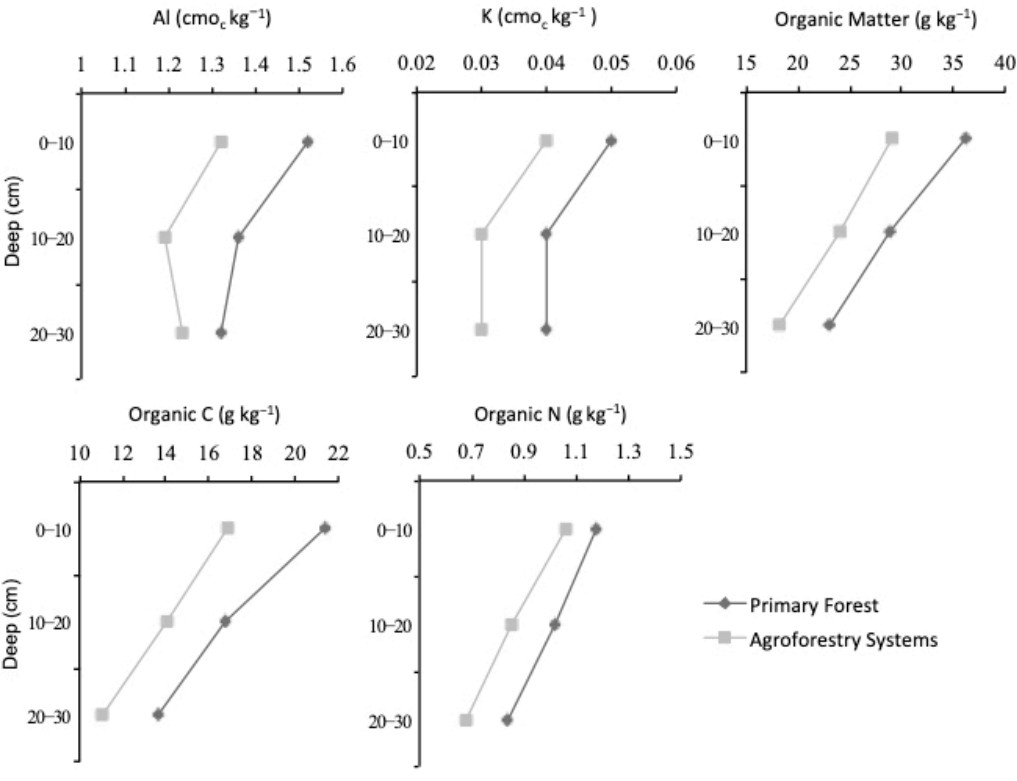

**Figure 2.** Aluminum, potassium, organic matter, organic carbon and nitrogen contents at three soil depths in the indigenous agroforestry systems of guarana and in adjacent forests ($n = 19$), in the Andirá–Marau Indigenous Land, Central Amazon region.

### 3.2. Biplot Analysis

A biplot is a graphical representation of the principal component analysis (PCA) in which the relations between levels of factors are observed, which permits a visualization of the interaction between the variables [21]. Three biplots were generated, one for the depth of 0–10 cm (Figure 3A), another for 10–20 cm (Figure 3B) and a third for 20–30 cm (Figure 3C). For each biplot, the chemical and granulometric characteristics of the soils of AFS and forest environments were used. The total variability "explained" by the PCA was 62.7% in layer A (0–10 cm), 63.9% in layer B (10–20) and 62.5% in layer C (20–30) (Figure 3).

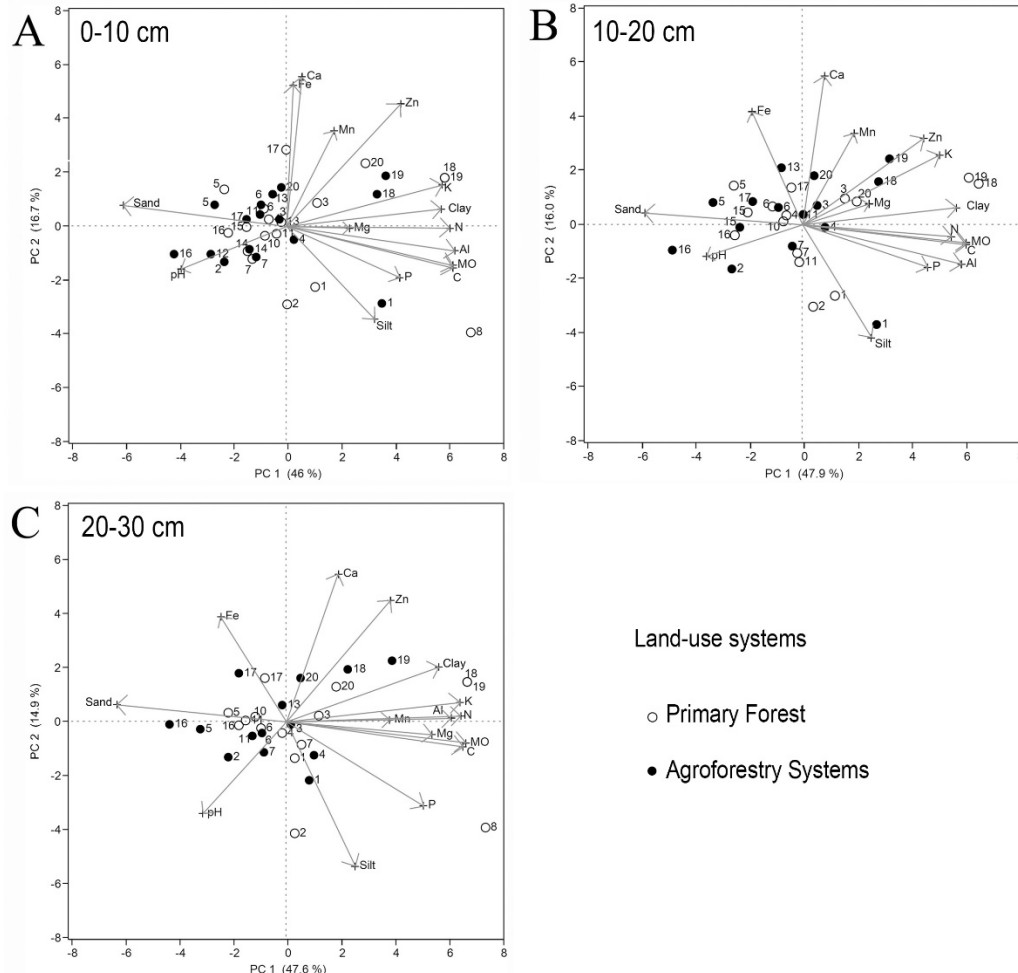

**Figure 3.** Biplots based on main components where the interaction between physicochemical traits of soil of AFS and forest for three soil depths is shown: 0–10 cm (**A**), 10–20 cm (**B**) and 20–30 cm (**C**).

## 4. Discussion

### 4.1. Soil Chemical Attributes

There were no significant differences in Ca, Mg and P between the soils of the two land-use systems sampled (Table 1), contrary to what has been observed in other studies in the Amazon, which showed increases in these nutrients in the soils of AFSs in relation to nearby forests or other natural vegetation [13,22,23]. However, these other studies examined homegarden agroforestry systems, where nutrient inputs from domestic residues can have a significant effect on soil fertility over time. The guarana AFSs, in contrast, are generally located at a considerable distance from dwellings, ranging from 100 m to over 3.6 km (mean = 1.09 km). While both the AFSs and the forests showed a soil pH somewhat above 4.0, the AFSs' pH was slightly higher, although that difference was not statistically significant. In both areas sampled, the Ca, Mg and P values were extremely

low, much lower than the critical levels (Ca < 0.4, Mg < 0.2, P < 3) suggested by other authors for agronomic production, indicating that these nutrients may limit production in the AFS [19,20]. Among the micronutrients analyzed, only Fe levels were considered high (>80 mg kg$^{-1}$), while the levels of Zn (<1.5 mg kg$^{-1}$) and Mn (<8 mg kg$^{-1}$) were far below the suggested minimums [19].

The main factor for increasing pH and reducing Al in AFSs could be the presence of ash resulting from previous use for slash-and-burn agriculture [13,22]. Guarana AFSs are established in two distinct cycles of land use: (1) through the enrichment of agricultural fields with guarana seedlings or (2) planting after the clearing of "capoeiras"—secondary vegetation approximately ten years old [9]. Similar soil results have been observed in AFSs in other regions of the Amazon such as Roraima [22], Amazonas [13,23] and Rondônia [24].

As most of the AFSs studied are more than twenty years old, the continuous exports of K and N in guarana crops over the years may have led to the gradual impoverishment of the soil in these nutrients. Moreover, the guarana plant is more demanding in N and K [3]. Associated with this, most AFSs are in soils with texture ranging from medium to sandy, in which the concentration of these nutrients tends to decrease due to leaching [25]. Ca, Mg, P and K values were also extremely low (<0.15 cmolc kg$^{-1}$) [19,20]. Nevertheless, a study on the export of macronutrients from guarana fruits in fields in the Maués region of Amazonas [3] observed that the amount of nutrients exported is low. This could explain how cultivation can continue even in soils with low levels of nutrient availability, as in the soils examined in the present study. The higher C and SOM content may be related to the greater vegetation cover and higher litter production in forest soils, as has been observed by other authors [13,22].

### 4.2. Biplot Analysis

The analysis shows the relationship between the presence of sand and low soil fertility. There was an agglomeration of the forest and the AFSs, and it was not possible to separate them, which shows that there is no difference between the sampled sites in regard to fertility, both being poor and sandy. Only two areas (numbers eighteen and nineteen) of the Monte Horeb community (Marau River) were associated with very clayey soils up to 30 cm deep, for both AFSs and forests, although they did not differ in terms of fertility [26]. It was also observed that the clay vector is almost in the same direction as the C, organic matter, N, P, K, Al and Mg vectors up to 30 cm deep. Therefore, clay is associated with the concentrations of these nutrients in the soil. However, at the pH levels observed, Fe and Al oxides also increase P fixation substantially [27]. A review of studies carried out in the Amazon Basin [28] found the C content in the soil to be strongly correlated with the clay content.

### 4.3. Organic Matter Management in the Guarana AFS

In general, the soils analyzed in this study have chemical characteristics with low values for plant nutrients, even in comparison to other soils of the Amazon region [13,22–24]. However, the maintenance of the levels of most nutrients in the AFSs, at the same level as observed in the forests, suggests that efficient nutrient cycling may be taking place. In the Amazon, despite the fragility of the soils, a complex network of interactions among plants, animals and microorganisms in the forest ensures efficient cycling of nutrients. Plants derive the bulk of their nutrition from the decomposition of plant material from other plants and from the decomposition of their own leaves [29]. In the conditions observed in this study, the management of organic matter in the AFS seems to be one of the main components contributing to the sustainability of production in these soils, considering that the levels of C, SOM and organic N were considered average [19,20], while the other chemical attributes analyzed, except for Fe, were always below the level considered minimum. The high efficiency in nutrient recycling observed in tropical forests has been correlated with high biological diversity. Among the macronutrients, N depends essentially on the internal recycling of the forest, and the fine litter represents its largest

input both to the forest ecosystem [30] and to agroforestry systems [31], with annual inputs much higher than those of the other macronutrients. Occasional losses can be compensated for by biological fixation of the legume components of the systems [29]. Organic C is one of the most important indicators used in assessing soil health [32] as it can improve its physical and chemical properties, increasing cation-exchange capacity (CEC) and recycling nutrients in addition to water retention capacity [33].

One of the traditional techniques used in the Sateré-Mawé AFS is the planting of the seedling in a "cradle" (Figure 4a), used to help protect the plant and favor root development, inhibit invasive herbs and maintain soil moisture. As the plant develops, the protection is removed or decomposes and contributes organic fertilizer. Decomposed tree trunk materials rich in nutrients [34], called "paú" by indigenous peoples and local communities, are added in the cradle (Figure 4b). During the first years, guarana needs a somewhat shaded environment, protected from wind and strong rain, and as the plant develops, shade is partially removed. In established guarana plantations, weeding and pruning are carried out between the months of February and May, and the organic matter is left on the soil. The old branches that have borne fruit are also pruned and left to decompose.

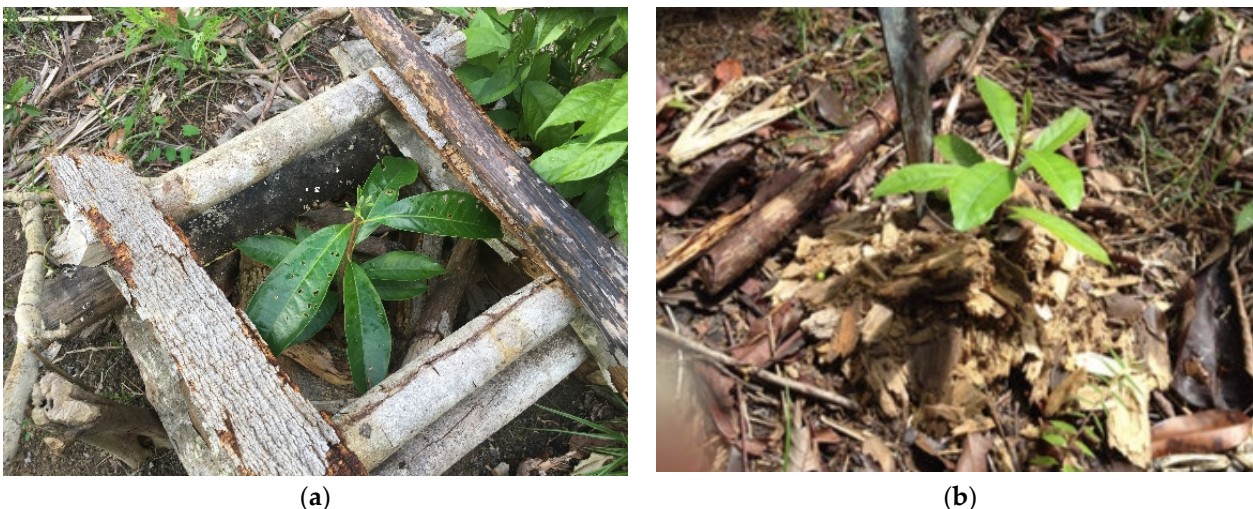

(**a**)  (**b**)

**Figure 4.** (**a**) "Cradle", a traditional practice of the Sateré-Mawé people for initial protection of seedlings in the field. (**b**) Decomposition of the wood used in the "cradle" later serves as organic fertilizer for guarana plants.

In 100% of the AFS, the soil is covered with litter from trees and other plants. Under these conditions, litter production and decomposition are the main nutrient sources for crop production [30,31,35]. Striffler and Vignoli observed that the litter of the guarana AFS of the Marau River valley can compensate for the nutrient exports [36]. The contribution of leguminous plants (Fabaceae) to the system can be particularly important, through the addition of organic matter rich in N. An inventory of the trees in the Sateré-Mawé guarana AFS found sixteen species of tree legumes native to the Amazon [9]. Of these, five (*Cedrelinga cateniformis* (Ducke); *Inga edulis* Mart; *Ormosia grossa* Rudd; *Stryphnodendron guianense* (Aulb.) Benth.; *Chamaecrista apoucouita* (Aubl.) H.S. Irwin and Barneby) are capable of fixing N [37,38].

Besides the contribution of legume trees to N fixation, the high tree diversity that characterizes the guarana systems, with an average of 134 species of trees found in the plots (mean of 19 species per plot, ranging from 0 to 38) [9] may also have an effect on soil fertility. Review articles [21] and experiments [39] indicate that a mix of tree species can positively affect soil fertility and overall productivity through several mechanisms. These range from differential exploration of root zones and access to nutrients to differences in leaf litter content, structure and decomposition rates, contributing to a greater array of organisms in the soil food web.



The traditional management practices of Sateré-Mawé farmers have helped to maintain the production of guarana AFSs in soils with low natural fertility; however, the use of other agro-ecological practices, such as the addition of limestone, green manure and organic waste, can help improve soil fertility and increase yield. Organic matter could come from guarana fruits themselves. They are composed of a pulp that covers the seed, formed by the outer skin and aril, and by the seed, composed of a seed coat and endosperm. While the seeds are commercialized, the pulp remains on the property as waste and could be a source of nutrients for the guarana plantations, given that it contains 55.8% K, 27.7% P and 60% Ca exported during harvest [3].

## 5. Conclusions

The soils of the guarana AFS and the surrounding forest have low natural fertility. Nonetheless, indigenous farmers have developed an agricultural system appropriate to local conditions, with the maintenance of average nutrient contents in the soil at levels similar to that of the adjacent forest, suggesting that efficient nutrient cycling is taking place. Under these conditions, management of organic matter seems to be one of the main factors in maintaining the productivity of guarana AFSs, along with the positive effects of the high tree diversity that indigenous farmers maintain as a part of the system. In contrast to the other soil attributes analyzed, which in general were below the level considered minimum for agricultural production, the levels of C, SOM and organic N remained at levels considered sufficient. The addition of limestone, green manures and organic residues from guarana processing, among other sources, can constitute low-cost alternatives to improve soil fertility and increase guarana production in the studied AFSs.

**Author Contributions:** Conceptualization, C.P.V., J.L., R.P.M. and S.S.A.; methodology, C.P.V., J.L., R.P.M., B.V.d.S., B.S. and S.S.A.; software, C.A.T.-B.; validation, C.P.V., J.L., R.P.M. and S.S.A.; formal analysis, C.A.T.-B.; investigation, C.P.V., J.L., R.P.M. and S.S.A.; resources, C.P.V., J.L., R.P.M. and S.S.A.; data curation, C.P.V.; writing—original draft preparation, C.P.V., J.L., R.P.M. and S.S.A.; writing—review and editing, C.P.V., J.L., R.P.M., C.A.T.-B., J.G.F.N. and S.S.A.; project administration, S.S.A.; funding acquisition, S.S.A. All authors have read and agreed to the published version of the manuscript.

**Funding:** This research was funded by the Petrobras Socioenvironmental Program, grant number 6000.0087008.13.2, with the objective of supporting sustainable productive activities focused on agroecology and environmental preservation.

**Institutional Review Board Statement:** The study was conducted according to the guidelines of the Declaration of Helsinki, and approved by the National Commission for Ethics in Research—CONEP (CAAE: 50259215.4.0000.0006), and the National Indian Foundation—FUNAI (Process: 08620.045951/2015-15). As this research involve access to traditional knowledge and entry into Indigenous Territory, authorizations were obtained from the National Institute Historical and Artistic Heritage—IPHAN (Authorization: 01450.005971/2015-03).

**Informed Consent Statement:** Not applicable.

**Data Availability Statement:** The data from this study are part of Clara Vignolli's Master Dissertation, which is available online: https://bdtd.inpa.gov.br/handle/tede/2621.

**Acknowledgments:** The authors thank the Sateré-Mawé Producers Consortium (CPSM) and the indigenous farmers of TI Andirá–Marau, who kindly contributed to this study. Amazonas State Research Support Foundation—FAPEAM, POSGRAD PROJECT 2021 and 2022.

**Conflicts of Interest:** The authors declare no conflict of interest.

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
