# Peer review of "Soil Management in Indigenous Agroforestry Systems of Guarana (Paullinia cupana Kunth) of the Sateré-Mawé Ethnic Group, in the Lower Amazon River Region"

_sustainability, doi:10.3390/su142215464_

Round 1

Reviewer 1 Report

  1. In this paper, the resolution of graphics needs to be improved, and some legends can not be distinguished.
  2. Please pay attention to the superscript and subscript in the article.
  3. The format of references should be unified, such as italics, page numbers, abbreviations, punctuation, authors, etc.
  4. 'efficient nutrient cycling' should be deep discussed. 
  5. Soil fertility in nitrogen fixation plants should be compared with those without such plants.

Author Response

Dear reviewer,
We forwarded the revised manuscript. In accordance with your suggestions, we inform you that the English language has been revised. The figures and legends were improved, as well as the format of the references.
Efficient nutrient cycling was more discussed, as well as nitrogen fixation and the manuscript was more adequately referenced.

Reviewer 2 Report

please see comments below:

Line 56 reference needed. Is biodiversity of native species comparable to undisturbed or unmanaged forest?

Line 83 is 0-10 started at the mineral Horizon or does this include the LFH? We’re soil Horizons classified? Line 90: this reference (16) is not available for me. Please describe in  greater detail. Line 92: Why was a ratio of 1:2.5 chosen? Line 106-108: was day tested for normality? Line 115-116: Language? Figure 2 seems to suggest that there increased organic C and organic matter storage in the primary forest. This difference is not reflected in the abstract. Why? Figure 3: Can you provide PCA without texture? Line 149-151 Language  Line 158: are the levels suggested by Cochrane relevant for the studied system? Please explain. A comparison to other systems assessed with the same methodology might be more relevant. Line 162: the difference in pH is minimal. What was the accuracy of the instrument used? Potentially differences in pH should not be discussed. Line 181- to 195: PCA seems to be non conclusive. Consider adding ellipse for group. Line 198-200: Please explain how e.g.  lower N suggests effective nutrient cycling. There seems to be a trend for a decrease. Line 196-208: References needed - please add where missing Line 214: which nutrients are released from the decomposing wood?      

Author Response

Dear reviewer,
We enclose the revised manuscript.
Thanks for the suggestions.

Response Reviewer 2

Line 56 reference needed.

  • References added

Porro, R.; Miller, R. P.; Tito, M. R.; Donovan, J. A.; Vivan, J. L.; Trancoso, R.; van Kanten, R. F.; Grijalva, J. E.; Ramirez, B. L.; Gonçalves, A. L. In Agroforestry in the Amazon region: a pathway for balancing conservation and development; Nair, P.K.R. (Ed.) Agroforestry—The Future of Global Land Use. Dordrecht: Springer, 2012, pp 391–428. DOI: 10.1007/978-94-007-4676-3_20

Is biodiversity of native species comparable to undisturbed or unmanaged forest?

  • No, there was no comparison with the diversity index in unmanaged forests. Species diversity of the agroforestry systems was compared using the Shannon index and with other agroforestry in the Amazon biome. Both the studied system and the ones used for comparison are composed mostly of native species, along with naturalized and exotic species.

Line 83 is 0-10 started at the mineral Horizon or does this include the LFH? We’re soil Horizons classified?

  • Soil horizons have not been classified. Sampling in the 0-10 cm layer was carried out by removing all the litter not decomposed or slightly decomposed on the soil surface.

Line 90: this reference (16) is not available for me. Please describe in greater detail. 

  • References added

Line 92: Why was a ratio of 1:2.5 chosen? 

  • The pH was determined in H2 This relationship was improved and standardized for tropical soils, according to crop responses to the application of inputs, correctives and fertilizers.

Line 106-108: was day tested for normality? 

  • It is unnecessary to test normality in PCA analysis because it is a descriptive analysis reducing n-dimensions to two dimensions. As traits have different scales, it is necessary to standardize each trait so that they are comparable.

Line 115-116: Language? Figure 2 seems to suggest that there increased organic C and organic matter storage in the primary forest. This difference is not reflected in the abstract. Why? 

  • As suggested, added to the abstract.

Figure 3: Can you provide PCA without texture? 

  • If you remove "sand, clay and silt", the discussion becomes meaningless. Some nutrients are related to sandy soils and others to clay soils. I should also add that the biplot is explaining about 60% of the total variation, so the reliability of the interpretations is high.

Line 149-151 Language  

  • add-on added

Line 158: are the levels suggested by Cochrane relevant for the studied system? Please explain. A comparison to other systems assessed with the same methodology might be more relevant. 

  • Information on the fertility status of Amazonian soils are limited. The work by Cochrane et al. is a guide to soils for agriculture in the Amazon, using laboratory methods similar to those used in the present work. A more recent reference was added, with specific data for soils in the state of Amazonas, also using the same methodology.
  • Moreira, A.; Fageria, N.K. (2009) Soil chemical attributes of Amazonas State, Brazil. Commun. Soil Sci. Plant Anal. 2009, 40, 1–14. doi:10.1080/00103620903175371

Line 162: the difference in pH is minimal. What was the accuracy of the instrument used? Potentially differences in pH should not be discussed. 

  • pH readings were taken in a potentiometer. More information was added in Materials and Methods.

Line 181- to 195: PCA seems to be non conclusive. Consider adding ellipse for group. 

  • "it was not possible to separate them" (forest and AFS). Therefore, the figure sent doesn’t have ellipse.

Line 198-200: Please explain how e.g.  lower N suggests effective nutrient cycling. There seems to be a trend for a decrease. 

  • Unlike the P, K, Ca and Mg contents, the N content was considered medium. Among the macronutrients, N depends essentially on the internal recycling of the forest, while the fine litter represents its major entry into both the forest ecosystem and agroforestry systems. The high contents of recycled N from the litter keeps this nutrient in the systems. In a forest in Central Amazonia, Luizão (1989) observed that with the fine litter the annual inputs of N were 151 kg ha-1, while for P, K, Ca and Mg the amounts were, respectively, 3, 15, 37 and 14 kg ha-1. On the other hand, Villani (2009) observed stocks of 217 kg ha-1 of N in the litter in indigenous SAFs of Western Amazonia, and 5.90, 54.2, 35.7 and 3.20 kg ha-1, respectively, of P, K, Ca and Mg. Recycling, avoiding leaching, represents a conservation mechanism for these nutrients in the system. Occasional losses can be compensated by biological fixation of the numerous leguminous components of the systems.
  • On this topic, two references were added in the manuscript.
  • Luizão, F.J. Litter production and mineral element input to the forest floor in a Central Amazonian forest. GeoJournal, 1989, 19, 407–417. https://doi-org.ez44.periodicos.capes.gov.br/10.1007/BF00176910
  • Villani, F.T. Dinâmica da matéria orgânica e fertilidade do solo em sistemas agroflorestais e outras coberturas vegetais em comunidades indígenas do Alto Solimões, Amazonas. Doctoral Thesis, National Institute of Amazonian Research, Manaus, AM, Brazil, 2009, pp. 128. https://repositorio.inpa.gov.br/handle/1/4975

Line 196-208: References needed - please add where missing

  • References added

Line 214: which nutrients are released from the decomposing wood?

Fernandes et al. (1997) conservatively estimated that removals of umburned wood debris can result in a loss from the site of around 380 kg N, 8 kg P, 92 kg K 61 kg Mg and 122 kg Ca per hectare. The derivation of these values is based on the assumption that: 1) 80% of above-ground woody biomass (say 300 Mg ha-1) is formed of trunks and coarse woody branches; 2) 40% of this biomass is consumed by fire; and 3) the nutrient contents of woody biomass are representative for "terra firme" forests.

On this topic a reference was added (line 226)

  • Fernandes, E.C.M.; Biot, Y.; Castilla, C.; Canto, A.C.; Matos, J.C.; Garcia, S.; Perin, R.; Wandelli, E. 1997. The impact of selective logging and forest conversion for subsistence agriculture and pastures on terrestrial nutrient dynamics in the Amazon. Cult. 1997, 49, 34-46. https://www.embrapa.br/busca-de-publicacoes/-/publicacao/1118883/the-impact-of-selective-logging-and-forest-conversion-for-subsistence-agriculture-and-pastures-on-terrestrial-nutrient-dynamics-in-the-amazon

Reviewer 3 Report

The article provides new insights to the readers of the journal

Sustainability. The article will have good impact to agroforestry research.

The manuscript is well organized and it is acceptable in its present form.

Author Response

Dear reviewer,
We enclose the revised manuscript.
Thanks for the suggestions.

Sonia Alfaia

Round 2

Reviewer 1 Report

The manuscript has been well revised and can be accepted as it is.